# Information Theory in Computational Biology: Where We Stand Today

**DOI:** 10.3390/e22060627

**Published:** 2020-06-06

**Authors:** Pritam Chanda, Eduardo Costa, Jie Hu, Shravan Sukumar, John Van Hemert, Rasna Walia

**Affiliations:** 1Corteva Agriscience™, Indianapolis, IN 46268, USA; jie.hu@corteva.com (J.H.); shravan.sukumar@corteva.com (S.S.); 2Computer and Information Science, Indiana University-Purdue University, Indianapolis, IN 46202, USA; 3Corteva Agriscience™, Mogi Mirim, Sao Paulo 13801-540, Brazil; eduardo.costa@corteva.com; 4Corteva Agriscience™, Johnston, IA 50131, USA; rasna.walia@corteva.com

**Keywords:** information theory, entropy, computational biology, gene expression, transcriptomics, sequence comparison, error correction, disease-gene association mapping, metabolic networks, metabolomics, protein structure, interaction analysis

## Abstract

“A Mathematical Theory of Communication” was published in 1948 by Claude Shannon to address the problems in the field of data compression and communication over (noisy) communication channels. Since then, the concepts and ideas developed in Shannon’s work have formed the basis of information theory, a cornerstone of statistical learning and inference, and has been playing a key role in disciplines such as physics and thermodynamics, probability and statistics, computational sciences and biological sciences. In this article we review the basic information theory based concepts and describe their key applications in multiple major areas of research in computational biology—gene expression and transcriptomics, alignment-free sequence comparison, sequencing and error correction, genome-wide disease-gene association mapping, metabolic networks and metabolomics, and protein sequence, structure and interaction analysis.

## 1. Introduction

Information theory has its roots in communication systems that deal with data compression and coding theorems for transmission of information from one source to another over noisy channels. Claude Shannon’s seminal work in communication theory from 1948 [1] provided the mathematical foundations for quantification and representation of information that made today’s digital era possible. It introduced the concept of channel capacity, defining the amount of information that can be sent over a noisy channel, bounded by the maximum possible transmission rate—“Shannon’s Limit” and stated that it is possible to transmit information through a noisy channel at a rate less than the maximum channel capacity keeping the probability of error at the receiver’s end arbitrarily small [2]. Since its inception, the significance and potentials of Shannon’s results were rapidly recognized, leading to further developments in probabilistic information measures and its wide application in diverse theoretical and applied fields outside of its original scope and intent in communication theory. Indeed, quantification and reduction of noise and uncertainty from data observations is key to improving statistical inference and it is not just limited to communication theory but any area that deals with noisy data. Biology is perhaps one of the experimental areas that is permeated by noise and variability in all observational levels, ranging from the most basic molecular and subcellular processes such as gene expressions and signaling pathways to complex interactions and dynamics of tissues, organs, organisms and populations [3]. Therefore, it is not surprising that information theory has found many theoretical advancements and witnessed myriad applications dealing with biological data. This is particularly true in the umbrella field of computational biology and bioinformatics that deals with computational applications of mathematical and statistical methods in the study of biological systems and processes. In this domain, information theory is widely used for model development and data analysis for a variety of biologically derived data types ranging from molecular, sequence and phenotypic data in genomics and genetics to gene expression, protein and spectral data in transcriptomics, proteomics and metabolomics, respectively [4,5,6,7,8,9,10,11].

Despite the numerous applications of information theoretic methods in many areas within computational biology, there have been only a few recent articles comprehensively reviewing theoretical developments and methodological applications to address problems in biology. Applications of information theory at the molecular level were reviewed in 2010 in [12] where the authors investigated information theory in the context of quantifying information in DNA-binding sites and the study of protein-DNA interactions and showed how coding theory can be used to explain molecular efficiency. In 2014, Vinga et al. [6] reviewed the application of information theory focusing exclusively on biological sequences. Their article discussed both global sequence analysis methods using metrics such as block-entropy (or L-tuple entropy) [13,14,15,16] that uses chaos game representation of genomic sequences [17,18], and local sequence analysis techniques such as classification of motifs, prediction of transcription factor binding sites and sequence characterization inspired by information theoretic frameworks with natural language underpinnings. More recently in 2016, information theory based methods to understand and quantify signal transmission in cellular systems were reviewed in [5]. Entropy based analysis has been widely applied in characterizing cell signal transduction and associated biochemical pathways that are inherently noisy owing to many factors such as the stochastic fluctuations in genetic circuits and intrinsic promiscuity of protein–protein interactions [19]. A review of multiple information theory based methods to quantify information processing at the level of individual cells in the context of biochemical networks with non-linear dynamics can be found in [5,7]. The article [5] also provided an excellent summary of entropy based methods investigating protein-interaction networks, while an earlier review authored by the same group [4] discussed information theory usage in gene regulatory and metabolic networks.

In this review article, we aim for the following—(1) we revisit some topics from previous reviews covering key newer entropy and information theory based approaches in those areas (gene regulatory networks, protein–protein interaction and metabolic networks); (2) we discuss information theory based measures of multivariate gene–gene interactions and survey articles using them in genome-wide disease-gene association mapping; (3) we offer a broad summary of information theory based applications by including discussion on several key and recent uses of information theory in topics within computational biology that were not collectively reviewed before (such as protein structure and interaction analysis, protein coevolution, sequencing error correction, alignment-free phylogeny, optimization and dimensionality reduction in biology).

The remainder of the article is organized into two major parts. In the first part, we introduce the readers to basic concepts, quantities and notations in information theory that form the basis of many advanced metrics and information theoretic algorithms discussed later. In the second part, we introduce and elaborate upon key applications of information theory in specific areas within computational biology mentioned in the preceding paragraph, focusing on more recent developments. Any application specific metrics and extensions and generalizations of the basic concepts will be discussed within the context of each application. Because of its mathematical nature, the first part will help to provide a uniform vocabulary and mathematical symbols to explain the applications discussed in the second part of the article.

## 2. Basic Metrics in Information Theory

### 2.1. Self-Information and Entropy

Shannon’s entropy [2] constitutes the basic building block for the information theoretic metrics to be discussed in the article. It is easiest to comprehend entropy when described in the context of uncertainty in discrete random variables. The primary inspiration behind entropy is based on the idea of information content of a probabilistic event. An event that occurs with high probability has less information content than an event that occurs with lesser probability. So, learning the occurrence of a less likely event such as “solar eclipse today” is more informative than knowing about a more likely event “rainy today”. This idea can be formally described using the metric of self-information (SI) that obeys the intuition of information content of an event. Given a discrete random variable *X* that assumes values from the set $V_{X}=\left\{x_{1},x_{2},\ldots ,x_{N}\right\}$ and follows a probability distribution $P_{X}$, SI for a single event $X=x_{i}$ is defined as
(1)$$\begin{array}{c} SI_{X}\left(x_{i}\right)=-\log\left[P_{X}\left(x_{i}\right)\right] \end{array}$$

Clearly, if an event $X=x_{i}$ occurs with certainty ($P_{X}\left(x_{i}\right)=1)$), $SI_{X}\left(x_{i}\right)=0$. This definition can be extended to an event pair involving two discrete random variables *X* and *Y*. For the joint occurrence of the events $X=x_{i}$ and $Y=y_{j}$ as:(2)$$\begin{array}{c} SI_{X,Y}\left(x_{i},y_{j}\right)=-\log\left[P_{X,Y}\left(x_{i},y_{j}\right)\right] \end{array}$$
where *Y* assumes values from the set $V_{Y}=\left\{y_{1},y_{2},\ldots ,y_{N}\right\}$, follows a probability distribution $P_{Y}$, and $P_{X,Y}\left(x_{i},y_{j}\right)$ is the joint probability of the two events $X=x_{i}$ and $Y=y_{j}$. This also highlights that when events described by $X=x_{i}$ and $Y=y_{j}$ are independent, the joint distribution should factor as $P_{X,Y}\left(x_{i},y_{j}\right)$ = $P_{X}\left(x_{i}\right)P_{Y}\left(y_{j}\right)$, so that the self-information $SI_{X,Y}\left(x_{i},y_{j}\right)$ = $SI_{X}\left(x_{i}\right)+SI_{Y}\left(y_{j}\right)$ is additive. Entropy (denoted mathematically by the symbol *H*) builds on the concept of self-information for a single event. For random variable *X*, it formalizes the information content of all the events $X=x_{i}$ (i=1,…,N) by taking the expectation with respect to the probability distribution of *X*:(3)$$\begin{array}{c} H\left(X\right)=-\sum_{x_{i}\in V_{X}}P_{X}\left(x_{i}\right)\log\left[P_{X}\left(x_{i}\right)\right]=E_{X}\left(-\log\left[P_{X}\right]\right) \end{array}$$

Entropy, thus, represents the uncertainty in *X* or information gained by observing a random variable *X* following the distribution $P_{X}$. Because it is an expectation, it depends on the distribution of the random variable *X* rather than an observed value. In other words, it represents the expected amount of information in an event occurrence following the distribution $P_{X}$ and provides a lower bound to the number of bits required (when log is base 2) on average to encode each event drawn from the distribution. If some events are much more likely than others, entropy will be low. For distributions that are close to uniform, entropy is high as uncertainty is high. Analogous definition exists for a continuous random variable *X*, where the entropy (also referred to as cross-entropy or differential entropy) is defined as,
(4)$$\begin{array}{c} H\left(X\right)=-\int P_{X}\left(x_{i}\right)\log\left[P_{X}\left(x_{i}\right)\right]dx_{i} \end{array}$$
where *P* refers to the continuous probability distribution of *X*.

As an example, consider *X* to be a random variable representing the alleles of a single nucleotide polymorphism (SNP) in the human genome. Assuming the SNP is bi-allelic, it can have 0, 1 or 2 copies of the minor allele in a genome. So *X* can take values from set {0,1,2} representing the number of copies of the minor allele in a genome at that position. If we identify the minor allele counts for the SNP across *N* genomes, $P_{X}\left(x=i\right)$, i∈{0,1,2} will give the respective probabilities for each of the three minor allele counts, and can be computed as $P_{X}\left(x=i\right)=n_{i}/N$, where $n_{i}$ is the frequency of minor allele count *i* and *N* is the total number of genomes. Then the entropy can be computed as $H\left(X\right)=-\sum_{i\in \{0,1,2\}}P_{X}\left(x=i\right)\log\left[P_{X}\left(x=i\right)\right]$. It will be zero when the SNP has a single allele; it is maximized when $P_{X}\left(x\right)$ has a uniform distribution, i.e., $P_{X}\left(x=i\right)=1/3$, i∈{0,1,2}.

The above definitions can be extended to multiple random variables. With a slight abuse of notation, for multiple discrete random variables $X_{1}$,&,$X_{M}$, entropy is defined using their joint probability distribution $P_{X_{1},\ldots ,X_{M}}$ as,
(5)$$\begin{array}{c} H\left(X_{1},\ldots ,X_{M}\right)=-\sum_{x_{1}\in V_{X_{1}}}\ldots \sum_{x_{M}\in V_{X_{M}}}P_{X_{1},\ldots X_{M}}\left(x_{1},\ldots ,x_{M}\right)\log\left[P_{X_{1},\ldots X_{M}}\left(x_{1},\ldots ,x_{M}\right)\right] \end{array}$$

### 2.2. Conditional Entropy

Given two random variables *X* and *Y*, the entropy of *X* given the event $Y=y_{i}$ is defined $H(X|Y=y_{i})$
$=-\sum_{x_{k}\in V_{X}}P_{X|Y}\left(x_{k}|y_{i}\right)\log\left[P_{X|Y}\left(x_{k}|y_{i}\right)\right]$$=E_{X|Y}\left(-\log\left[P_{X|Y}\right]\right)$ from which the conditional entropy of *X* given *Y* is defined as [2]
(6)$$\begin{array}{c} H\left(X|Y\right)=\sum_{y_{i}\in V_{Y}}P_{Y}\left(y_{i}\right)H\left(X|Y=y_{i}\right)=E_{Y}\left(E_{X|Y}\left(-\log\left[P_{X|Y}\right]\right)\right) \end{array}$$

Continuing with the example of the SNP from above, assume *X* is the SNP random variable and *Y* is a random variable associated with a measured phenotype, say presence (Y=1) or absence (Y=0) of a diseased condition. Assume $n_{ij}$ be the frequency of minor allele count i∈{0,1,2} for all genomes with a given disease status Y=j, j∈{0,1}. Also assume $c_{j}$ be the frequency of disease status Y=j and *N* be the total number of genomes. Then the empirical conditional probability distribution is given by $P_{X|Y}\left(X=i|Y=j\right)=n_{ij}/c_{j}$; the conditional entropy can be calculated as,
$$\begin{array}{cc} H(X|Y) & =\sum_{j\in \{0,1\}}P_{Y}\left(Y=j\right)H\left(X|Y=j\right)\\ & =-\sum_{j\in \{0,1\}}P_{Y}\left(Y=j\right)\sum_{i\in \{0,1,2\}}P_{X|Y}\left(X=i|Y=j\right)log\left[P_{X|Y}\left(X=i|Y=j\right)\right] \end{array}$$

The marginal distribution of *Y* is given by $P_{Y}\left(Y=j\right)=c_{j}/N$.

### 2.3. Relative Entropy

While Shannon’s definition of entropy provides the average information content in a given probability distribution, it is often necessary to obtain information content of the occurrence of events in a probability distribution $P_{X}$ with respect to a reference probability distribution $Q_{X}$, which is given by relative entropy or Kullback–Leibler Divergence [2,20]. It can be thought of as a measure of the distance between two distributions and measures the loss in information of using probability distribution $Q_{X}$ when the true distribution is $P_{X}$ and is given by
(7)$$\begin{array}{c} KLD(P_{X}\left|\right|Q_{X})=\sum_{x_{i}\in V_{X}}P_{X}\left(x_{i}\right)\log\left[\frac{P_{X}\left(x_{i}\right)}{Q_{X}\left(x_{i}\right)}\right]=E_{X}\left(\log\left[\frac{P_{X}}{Q_{X}}\right]\right) \end{array}$$
where the expectation is with respect to the true density $P_{X}$. There are many powerful mathematical relationships that relate the information theoretic KLD directly to concepts and ideas from statistics. For example, the KLD is the expected log-likelihood ratio and the $\chi^{2}$ statistic is twice the first term in the Taylor expansion of the KLD [21].

For random variables with continuous distributions, KLD is given by,
(8)$$\begin{array}{c} KLD(P_{X}\left|\right|Q_{X})=\int P_{X}\left(x\right)\log\left[\frac{P_{X}\left(x\right)}{Q_{X}\left(x\right)}\right]dx \end{array}$$
where the integration is over the domain of *X*. The KLD is asymmetric (i.e., $KLD(P_{X}\left|\right|Q_{X})\neq KLD(Q_{X}\left|\right|P_{X})$ when $P_{X}\neq Q_{X}$), always takes non-negative values, additive, and is zero only if $P_{X}$ = $Q_{X}$. To obtain a more symmetric measure, the Jensen–Shannon divergence (JSD) [22] is defined as
(9)$$\begin{array}{c} JSD\left(P_{X},Q_{X}\right)=\frac{1}{2}KLD(P_{X}\left|\right|M_{X})+\frac{1}{2}KLD(Q_{X}\left|\right|M_{X}) \end{array}$$
where $M_{X}=\frac{1}{2}\left(P_{X}+Q_{X}\right)$. JSD is symmetric and bounded between 0 and 1.

### 2.4. Mutual Information

The mutual information of two random variables *X* and *Y*, denoted by MI(X;Y) describes the information in probability distribution of one random variable *X* about the distribution of another random variable *Y* [2,20]. It is defined as the reduction in uncertainty in the random variable, say *X* after observing the other, *Y*, which can be shown to be equivalent to the relative entropy of the joint distribution of *X* and *Y*, P(X,Y) relative to the product of their marginals, P(X)P(Y):(10)$$\begin{array}{c} MI\left(X;Y\right)=\sum_{x_{i}\in V_{X}}\sum_{y_{i}\in V_{Y}}P_{X,Y}\left(x_{i},y_{i}\right)\log\left[\frac{P_{X,Y}\left(x_{i},y_{i}\right)}{P_{X}\left(x_{i}\right)P_{Y}\left(y_{i}\right)}\right]=KLD(P_{X,Y}\left|\right|P_{X}P_{Y}) \end{array}$$

MI provides a measure of association or correlation between *X* and *Y* and is often denoted as “interaction information” between the two random variables. MI reflects the reduction in uncertainty (i.e., the information gained) for *Y* when *X* is known. Using the definition of entropy and conditional entropy, MI(X,Y) can be expressed as
(11)$$\begin{array}{c} MI(X;Y)=H(X)+H(Y)-H(X,Y)=H(X)-H(X|Y)=H(Y)-H(Y|X) \end{array}$$

Using the example of the SNP *X* and disease phenotype *Y* from above, MI between the *X* and *Y* can be easily calculated from the individual entropies H(X), H(Y) and the joint entropy H(X,Y). H(X) can be given by $H\left(X\right)=-\sum_{i\in \{0,1,2\}}P_{X}\left(x=i\right)\log\left[P_{X}\left(x=i\right)\right]$ with $P_{X}\left(x=i\right)=n_{i}/N$; H(Y) is computed as $H\left(Y\right)=-\sum_{j\in \{0,1\}}P_{Y}\left(y=j\right)\log\left[P_{Y}\left(y=j\right)\right]$ with $P_{Y}\left(y=j\right)=c_{j}/N$. Here, $n_{i}$ is the frequency of minor allele count *i*, $c_{j}$ is the count of genomes with disease phenotype *j* and *N* is the total number of genomes. The joint entropy H(X,Y) can be obtained as $H\left(X,Y\right)=-\sum_{i\in \{0,1,2\}}\sum_{j\in \{0,1\}}P_{X,Y}\left(x=i,y=j\right)\log\left[P_{X,Y}\left(x=i,y=j\right)\right]$. The joint distribution can be estimated as $P_{X,Y}\left(x=i,y=j\right)=n_{ij}/N$, where $n_{ij}$ is the count of genomes with observed minor allele count *i* and disease phenotype *j*.

### 2.5. Interaction Information

The above definition of MI as a measure of interaction between two random variables can be extended to *k* variables using a multivariate generalized definition of interaction information also known as *k*-way interaction information (or KWII). It is defined as the amount of information (synergy or redundancy) that is present in the set of random variables, which is not present in any subset of these variables [23,24]. For a set of *k* random variables $S=\{X_{1},X_{2},\cdots X_{k}\}$, the KWII can be written succinctly as an alternating sum of the entropies of all possible subsets τ⊆S using the difference operator notation of Han [25]:(12)$$\begin{array}{c} KWII\left(X_{1},\cdots X_{k}\right)=-\sum_{\tau \in S}{\left(-1\right)}^{\left|S-\tau \right|}H\left(\tau \right) \end{array}$$

As an aid to understanding the above equation, for the simple case of *k* = 3 discrete random variables, the KWII is given by
(13)$$\begin{array}{c} KWII\left(X_{1};X_{2};X_{3}\right)=-H\left(X_{1}\right)-H\left(X_{2}\right)-H\left(X_{3}\right)+H\left(X_{1},X_{2}\right)+H\left(X_{2},X_{3}\right)+H\left(X_{1},X_{3}\right)-H\left(X_{1},X_{2},X_{3}\right) \end{array}$$

The value of KWII(S) can be both positive and negative where larger positive values indicate stronger interaction information (i.e., higher association) among the variables in *S*, while negative values indicate redundancy of information between the variables. As an example, for two SNP random variables $X_{1}$ and $X_{2}$ and disease phenotype random variable *Y*, the interaction information given by $KWII(X_{1};X_{2};Y)$ can be computed using entropies of all possible subsets as above. It can be interpreted as a measure of the synergistic association of the predictor variables $X_{1},X_{2}$ with *Y*, i.e., how well $X_{1}$,$X_{2}$ explains *Y*—positive values will indicate that interaction between $X_{1}$ and $X_{2}$ enhances prediction of *Y*, while negative values will indicate that the prediction is reduced or inhibited.

## 3. Applications of Information Theory in Computational Biology

### 3.1. Gene Expression and Transcriptomics

Identifying, modeling and characterizing gene–gene relationships and their co-expression patterns is an important challenge in molecular and systems biology, as these dynamic interactions determine cellular phenotypes and represent causality inherent in developmental processes and regulatory pathways. Such associations are typically modeled computationally and statistically ’reconstructed’ or ’reverse engineered’ using Gene-regulatory Networks (GRNs, also referred to as Transcriptional Regulatory Networks) utilizing gene expression and mRNA abundance data obtained using technologies such as microarrays and RNAseq. More recently, large expression data availability from single cell experiments has spurred developments of new computational methods to infer GRNs, model regulatory interactions and gain insights into cell fate decisions and associated transcriptional state changes [26,27,28,29,30]. A GRN is modeled as an undirected graph or a network where the genes (and transcription factors) are represented as vertices (or nodes) and are connected by edges representing regulatory interactions between transcription factors and their targets. The gene expression data matrix (experimental or synthetic) typically provides the genes in the rows and samples and experimental conditions in the columns. The goal of the network inference method is to use the expression matrix to infer the set of regulatory interactions (direct or indirect regulation) between any two genes in the GRN, thereby predicting the edges in the network (Figure 1). Because mutual information (MI) provides the ability to capture non-linear dependencies between two variables, several methods for reconstructing GRNs use MI or associated information theoretic scores to infer the regulatory relationships. As a first step, these methods require the computation of the MI Matrix (MIM), a square matrix in which the element at the *i*th row and *j*th column is given by the MI between genes $G_{i}$ and $G_{j}$. Often, this computational step has affordable complexity given that only pairs of MI computations based on bivariate probability distributions are needed to be estimated to complete the MIM. Several information theoretic methods have been developed to reconstruct GRNs over the past two decades [31,32,33]. For the remainder of this section, we will briefly review some of the popular information theory based methods that were discussed in-depth in some prior reviews [4,8] and then focus our discussion on more recent advancements in this area in the context of analysis of single-cell gene expression data.

One of the earliest approaches using MIM can be found in Relevance Networks [34] that simply links a pair of genes by an edge if the MI is larger than a given threshold. The approach has been introduced to infer relationships between RNA expressions and finding clusters of genes that affect cancer susceptibility to anticancer agents [35]. The Context Likelihood of Relatedness (CLR) algorithm [36] is an extension of the relevance network approach that calculates the statistical likelihood of a particular transcription factor/gene pair’s MI value within its network context and filters many of the false (noisy) edges in the constructed GRN where one gene can weakly co-vary with many transcription factors simply by chance due to inherent noise in biological systems. It compares the MI between the transcription factor/gene pair to the background distribution of MI values for all possible pairs that include either the transcription factor or its target gene, and retains only interactions that have MI scores significantly above the background distribution. Both Relevance Networks and CLR suffer from a fundamental limitation—genes in indirect relationships separated by one or more intermediate genes may be highly co-regulated, resulting in numerous false positives. To address this, ARACNE (Algorithm for the Reconstruction of Accurate Cellular Networks) [37], a popular algorithm for reconstructing gene networks, was developed. ARACNE also starts with estimating the MIM between the genes, but it then filters spurious gene–gene associations using a two step process. First, each MI value between a gene pair is filtered using an appropriate threshold computed from the null-hypothesis of independence between the two genes, the null distribution being obtained by randomly shuffling the gene expression values for the given pair of genes. In the second step, ARACNE removes the vast majority of indirect associations using the data processing inequality (DPI) information theoretic assertion that states that if there is an indirect association between two genes $G_{i}$ and $G_{j}$ through gene $G_{k}$, then $MI\left(G_{i},G_{j}\right)\leq \min\left\{MI\left(G_{i},G_{k}\right),MI\left(G_{k},G_{j}\right)\right\}$. It means that gene $G_{i}$ cannot have more information about gene $G_{j}$ than gene $G_{k}$ has about gene $G_{i}$ and the post-processing cannot increase information. More recent extensions of ARACNE can be found in [38,39,40]. Time-Delay-ARACNE is proposed by Zoppoli et al. for GRN inference from time-course expression data [38], while higher order DPI is used to improve inference performance in hARACNE [39]. Improved computational performance is reported by Lachmann et al. in ARACNE-AP [40] by using an adaptive binning strategy to estimate MI.

In the past few years, with the advancements in single-cell technologies for biological analysis, it has now become possible to quantify single cell transcriptomes in very large numbers. While that has made it possible to provide a finer-grained picture of the complex cellular processes and decipher heterogeneity inherent in gene expression across multiple tissue types, analysis of gene expression data from single cells comes with different data distribution patterns and characteristics (e.g., has a higher rate of zero values) often distinct from their bulk sample counterparts [8]. These present more complexities for statistical analysis (such as more technical noise and biological variability) [27]. As a result, GRN reconstruction methods developed for bulk sample expression data may not be suitable for data generated from single cells and may not take full advantage of the cell-to-cell variability to infer statistical relationships that can be used by information theory. Taking into consideration some of the different characteristics of single cell expression data, Chan et al. [27] developed a multivariate information theory based approach using partial information decomposition (PID) [41] computed from every triplet of genes in single-cell gene expression datasets. For three genes $G_{i}$, $G_{j}$, $G_{k}$, PID provides a decomposition of the information provided by two of the genes as source variables (say $S=\{G_{i},G_{j}\}$) about the third gene $G_{k}$ into three components—redundant, unique and synergystic information.
(14)$$\begin{array}{c} PID\left(S;G_{k}\right)=Synergy\left(G_{k};G_{i};G_{j}\right)+Unique_{G_{j}}\left(G_{k};G_{i}\right)+Unique_{G_{i}}\left(G_{k};G_{j}\right)+Redundancy\left(G_{k};G_{i};G_{j}\right) \end{array}$$

The inference algorithm developed in [27] (named PIDC or “PID and Context”) uses the Proportional Unique Contribution (PUC) metric defined as the average ratio of unique information between two genes $G_{i}$ and $G_{j}$ given the presence of a third gene $G_{k}$ from the remaining genes *S*:(15)$$\begin{array}{c} U_{G_{i},G_{j}}=\sum_{G_{k}\in S\setminus \left\{G_{i},G_{j}\right\}}\frac{Unique_{G_{k}}\left(G_{i};G_{j}\right)+Unique_{G_{k}}\left(G_{j};G_{i}\right)}{MI(G_{i};G_{j})} \end{array}$$

Next, the confidence of an edge is calculated as the sum of the cumulative distribution functions of all PUC scores for each gene in the pair:(16)$$\begin{array}{c} conf\left(G_{i},G_{j}\right)=F_{G_{i}}\left(U_{G_{i},G_{j}}\right)+F_{G_{j}}\left(U_{G_{j},G_{i}}\right) \end{array}$$
where $F_{G_{i}}\left(U\right)$ is the cumulative distribution function of all PUC scores involving gene $G_{i}$. Like CLR, estimating confidence of an edge by incorporating distributional properties of the PUC values for a pair of genes improves robustness to noise. It effectively identifies the most important interactions per gene, instead of relying on a simple ranking of the pairwise scores and choosing the best scoring edges across the whole network. Comparing PIDC with ARACNE and CLR, the authors reported marginal improvement in inferring the GRNs using simulated and publicly available data sets, attributing their success to larger sample sizes provided by single-cell data. However, it remains unclear as to how PIDC takes advantage of specific characteristics of single-cell data, such as its zero-rich nature.

Intrinsic noise in transcriptomic and gene expression data, owing to the stochastic nature of biochemical reactions driving the production of mRNAs and proteins within cells, makes GRN inference and reconstruction an extremely challenging problem. In addition to information theoretic methods, statistical methods such as Bayesian networks, tree based [42] and ordinary differential equation based [43] methods have been proposed to analyze gene expression data and reconstruct GRNs. An advantage of Bayesian networks and tree based methods is that they can learn directionality of the edges conjecturing if there is a directional influence from a regulatory gene to a target gene; information theoretic methods typically infer undirected edges. A head-to-head comparison of various methods to infer GRNs using single cell expression data has been presented recently in [8]. The methods were evaluated in their abilities to recover the correct set of GRN edges using ROC curves and Precision-Recall curves against reference networks from experimental assays, as well as in-silico reference networks from simulated data sets with single-cell characteristics. They reported that most of the assessed methods, including information theoretic methods, are not able to predict network structures from single cell expression data accurately and have differences with each other in the sets of identified edges. Also, their performances are inconsistent across data from different cell types and experimental conditions, as previously reported in another comparative study [44]. Both studies found no single best performing method across multiple types of data and simulation scenarios; rather, ensemble methods that average prediction using multiple approaches seem to work best [8,44]. The results from these studies emphasize the necessity to develop more accurate and optimized network modeling methods that are compatible with single cell expression data.

### 3.2. Alignment-Free Sequence Comparison

In this section, we discuss prominent applications of information theory for finding similarity between two genomic sequences without doing an actual alignment of the two genomes. Comparing two or more genomic sequences to find regions of similarities and dissimilarities and infer a measure of relatedness is vital for the success of basic phylogenetic and metagenomics research. Sequence comparison of genetic material across organisms allows discovery of gene structures and inferring their functional relationships based on the idea that higher similarity between sequences is a driver for higher structural and functional similarity and possible evolutionary relationships. While traditional methods for comparative sequence analysis and phylogeny reconstruction rely on computational algorithms for pairwise and multiple sequence alignments such as dynamic programming, they are often relatively slow in aligning two sequences, taking time proportional to the product of their lengths and are difficult to scale to whole genome comparisons. Furthermore, an underlying assumption supporting sequence alignment algorithms is the principle of co-linearity (homologous sequences have multiple linearly arranged and more or less conserved sequence stretches), which is often violated in biological genomes due to rearrangement events (e.g., in viral genomes) [45]. These and other challenges, such as a rapid drop in accuracy of sequence alignments when sequence identity falls below a certain critical point, have prompted development of alternative alignment-free methods for comparing large genomic sequences [46]. Among many different approaches to alignment-free sequence comparison (such as word frequency based methods inspired by linguistic analysis [47], graphical representation of DNA sequences [48,49], chaos game representation [50], iterated maps [51], spaced words [52,53]), applications of information theory have been key to developing methods utilizing the informational content between sequences to be compared. In the remainder of this section, we briefly review some of the key articles using information theory for alignment-free sequence comparison.

In the feature frequency profiles (FFP) method [54] developed by Sims et al., the authors proposed to estimate the distances between nucleotide or amino acid sequences using a two step process. In the first step, fixed length words (or k-mers) and their frequencies are extracted from each genomic sequence by sliding a window along the genome. Normalizing each frequency with the sum of all k-mer frequencies generates a probability distribution for the genome. This process leads to the conversion of each genomic sequence into its FFP, $F_{k}$, represented by the distribution of each word in the genome. In the second step, pairwise distance between two sequences *i* and *j* can be calculated using the Jensen–Shannon divergence (JSD) between their respective FFPs, $F_{i}$ and $F_{j}$. The JSD metric symmetricizes the asymmetric KLD distance metric between two probability distributions. The JSD serves as a measure of distance between the two genomes and can be used to generate phylogenetic trees from multiple genomes. The authors also used information theory to address a key question in their approach—how can one choose an optimal value of *k* (the word length). Choosing a large value of *k* will run into higher computational complexity and possibly unreliable estimates of k-mer probability distributions. Choosing a smaller value will result in words that are too small, commonly occurring in all genomes and does not have the information to distinguish between the genomes. To address this, the authors computed lower and upper bounds of word size *k*. The lower limit is given by the word length that is most frequent in a genome of interest. The upper limit is derived using the k−2 Markov model, which says that one can predict the frequency of a word of length *k* using the frequencies of its k−1 and k−2 subwords. So for a given word length *k*, the expected FFP, $\hat{F_{k}}$, can be computed and KLD or relative entropy is then calculated between $F_{k}$ and $\hat{F_{k}}$ for all values of *k* from one to infinity and summed to get CRE (cumulative relative frequency). The CRE represents the accuracy of predicting FFPs for all lengths ≥k, given the prior distributions $F_{k-1}$ and $F_{k-2}$. The value of *k* at which the CRE approaches 0 becomes the upper limit of word size for use in genome comparison.

More recently, probabilistic data structures for k-mer counting have been used to enable alignment-free genome comparison [55,56]. A critical advantage of these data structures is higher memory efficiency allowing them to be used for comparing many sample sequences at the same time using next generation sequence data. One of these methods, kWIP, hashes all k-mers from a genomic sequence *i* into a probabilistic data structure called a Sketch [56,57], $S_{i}$, that are numeric vectors of fixed size for every sequence. Given a set of many sequences (e.g., from a metagenomics experiment), genetic relatedness between any two sequences *i* and *j* is then calculated by computing an inner product between the two sketches $S_{i}$ and $S_{j}$, weighted by their informational entropy across the population set. This procedure down-weights uninformative k-mers (highly abundant or present in very few samples) and produces a kernel matrix, *K*, of pairwise inner products that is then normalized using the Euclidean norm ${K^{\prime }}_{ij}=K_{ij}/\left(K_{ii}K_{jj}\right)$ and converted to distance matrix $D_{ij}=\sqrt{{K^{\prime }}_{ii}+{K^{\prime }}_{jj}-2{K^{\prime }}_{ij}}$ containing the genetic distances between every pair of sequences in the population. A schematic illustrating the FFP and kWIP approaches is presented in Figure 2.

Over the past few years, many alignment-free algorithms have been proposed [58,59,60] to solve previously intractable challenges in phylogenomics, many of which have been compared in [9,45] using data from three categories—gene tree inference, genome-based phylogeny and horizontal gene transfer. Assessing the degree of topological disagreement between the inferred and reference trees, the report found that no single method performed best across all the data sets. When compared using eight data sets from the whole-genome phylogeny and horizontal gene transfer categories, FFP was found to be among the top five best performing methods out of 55 variants of 11 alignment-free phylogeny tools tested. These suggest that improved information theoretic distance metrics to assess similarities between genomes need to be developed as they were often more challenged in deciphering finer complex organization levels in the sequences [45]. With genomics continuing to lead in terms of data growth and availability through production of high-throughput sequencing data, although information theory based methods are observed to be relatively more memory efficient and computationally inexpensive than other methods for smaller data sets, bigger data sets pose more serious challenges, for example, substantial memory overhead when using longer k-mers. Analyses of very large data sets containing beyond tens of thousands of samples will benefit from more optimized implementations of methods such as kWIP that performs pairwise computations, including parallelization across distributed processing systems. Therefore, it remains a challenge in alignment-free phylogeny to develop both novel metrics and efficient implementations that can scale to handle larger data sets.

### 3.3. Sequencing and Error Correction

Information theory based approaches have notable applications in genome sequencing. DNA sequencing technologies have continuously evolved over the past two decades, leading to a substantial increase in throughput and decrease in cost. Current technologies are mainly based on a two-step approach. First, fragments of the target sequence/genome are read via shotgun sequencing. Then, these fragments, so-called reads, are assembled to reconstruct the original sequence. This reconstruction task can be seen as a problem of assembling a string from its substrings, where three aspects play an important role in the completion of the task: length of the fragments; number of reads per base (coverage depth); and fragment error rates. In the case of DNA sequencing, these aspects are highly dependent on the sequencing platform. Average read length typically ranges from 100 to 10,000 base pairs (bp), although platforms such as PacBio and Oxford Nanopore Technologies MinION can reach maximum read lengths in the order of 60–100 kbp [61,62]. The number of reads typically ranges from 100 s to 1000 s of millions. And the error rate can range from 0.1% to 20% [63,64].

Given a set of reads, the most crucial challenge for the genome reconstruction task is to determine whether complete reconstruction of the target sequence is possible from the given set of reads. Additionally, other important challenges include finding the minimum fragment length and coverage depth needed for the complete reconstruction with a given reliability, evaluating the impact of the read error rates on the assembly performance and on the requirements in terms of read length and coverage depth, and designing ways to compare different sequencing technologies and assembly methods. These challenges have been recently tackled from an information-theoretic perspective by investigating the fundamental limits of genome reconstruction. From an information theory point of view, a sequencing task can be seen as a decoding task where the genetic information is first encoded in terms of nucleotide bases and then transmitted during sequencing through a noisy channel in form of reads (Figure 3).

Motahari et al. [66] investigated the fundamental limits for the read length and the number of reads needed to allow complete sequence reconstruction, assuming error-free reads and DNA sequence modeled as an i.i.d (independent and identically distributed) random process. They calculated the minimum read length required for complete sequence reconstruction based on the Rényi entropy [67] of order two, which is defined as $H_{2}\left(p\right)=-\log_{2}\sum_{i}p_{i}^{2}$, where *p* is the probability distribution on the alphabet {A,C,G,T} and *i* iterates over each of these values. More specifically, given read length *L* and original sequence length *G*, they calculated the normalized read length $\bar{L}=L/\log_{2}\left(G\right)$ and showed that if $\bar{L}<2/H_{2}\left(p\right)$, complete sequence reconstruction is impossible, regardless of the number of reads. If read length is above that threshold, then complete reconstruction requires that enough reads exist to cover every base least once. These findings were later extended to consider scenarios that do not require the i.i.d DNA sequence assumption or reads to be error-free [68,69,70].

Gabrys et al. [71] studied these questions in the more general context of string reconstruction, motivated by applications in DNA-based data storage systems [72], discussed later in this section. They investigated the fundamental limits of read overlap and their impact on the read length requirements, and established the minimum Hamming distance between reads to allow proper sequence reconstruction. Marcovich and Yaakobi [73] followed the work in [71], by exploring two models for string reconstruction: one in which not all substrings are received, and another in which all substrings are received, but with errors. They also proposed substring constraints based on the Hamming distance between reads.

In another proposed method [74], Si et al. investigated the closely related haplotype assembly problem, where the goal is to reconstruct haplotype sequences (i.e., string of single nucleotide polymorphisms (SNPs) on a single chromosome in a homologous pair). Using an information-theoretic perspective, they formulated this goal to be the successful recovery of two sources of information being communicated through a channel: the haplotype information and the chromosome membership. They studied the required conditions to allow a reliable reconstruction with and without error-free reads and showed that the requirements in terms of number of reads for the erroneous case is of the same order as in the error-free case. These limits were defined using the Fano’s inequality [2].

More recently, in a novel approach, Chen et al. [65] combined sequencing-by-synthesis with an information theory-based error-correction algorithm, as outlined in Figure 3. They proposed a communication channel with three rounds of fluorogenic DNA sequencing [75] and use information theory principles to analyze information redundancy. In each round, the sequencer generates a dual-base array, which is combined in a later stage to infer the four-base DNA sequence. For example, one of the rounds generates the so-called *MK* dual-base array to determine whether the DNA base is *A* or *C* (encoded as *M*) or *G* or *T* (encoded as *K*) for every position of the sequence; in this case, *K* and *M* are called degenerate bases. Considering a random DNA sequence, the entropy of a sequence of length *L* is 2*L* bits, while the entropy of a dual-base array is *L* bits, assuming no sequencing error. Therefore, if two different dual-base arrays are used, the original sequence can be fully reconstructed. However, as the actual entropy of each dual-base array is lower than *L*, because of sequencing errors, a communication channel based on only two rounds, one for each dual-base coding, would be prone to propagate sequencing errors. Based on the experimental error rate of fluorogenic DNA sequencing, the authors showed that using three dual-base array provides enough information for sequence reconstruction with error correction. Besides the *MK* dual-base array, they used the *RY* array, where *R* means *A* or *G* and *Y* is *C* or *T*, and the *WS* array, where *W* means *A* or *T* and *S* is *C* or *G*. By combining the information encoded by the three degenerate arrays, the decoder can correct an error of one of the rounds. Experimental results in [65] showed that this strategy reduces the error rate in a 250 bp sequence from 0.96% to 0.33%. In a recent study, Mitchell et al. [76] benchmarked several computational error-correction methods for sequencing data and reported that Chen et al.’s method [65] suffered from increased computational cost compared with other methods that limits its scalability to larger data sets.

The same concept of degenerate bases is used in [77,78], but with the focus of increasing information capacity for DNA based data storage when storing results for multiples reads from a target sequence. By using different encoding bases (original and degenerate ones) to encode the observed base frequencies for each position of the sequenced reads, the need to store all reads is eliminated. As a finite alphabet of encoding bases, including pure and degenerate bases, the observed base frequency will most likely not match any of the bases in the alphabet. Anavy et al. [77] solved this problem by using KLD to make the best approximation, while Choi et al. [78] used a clustering approach. They both used Reed–Solomon coding [79] as a correction mechanism.

Improvements in error correction for long-read sequencing (LRS) technologies, such as PacBio and MinION, is one of the current challenges in DNA sequencing [80,81]. LRS-based approaches have shown advantages in sequencing complex regions, such as those with extreme GC-rich, long tandem repeats, and interspersed repeats, when compared to prevailing short-read approaches [82,83]. However, their high error rates in DNA sequencing are still considered a main drawback [84], and information theoretic methods have the capability to reduce it, as shown in [65]. Information theory offers the potential to further contribute in this area by defining fundamental limits for the coverage depth of DNA reads, to guarantee full reconstruction, error correction, and minimum redundant information. Using information theory for biology inspired and error-free information storage is another emerging area of research [72,85,86,87]. It has drawn more attention lately because of the novel use of DNA as an archival medium and holds a lot of promise for future research to satisfy the exponentially increasing demand for information storage.

### 3.4. Genome-Wide Disease-Gene Association Mapping

Over the last two decades, fueled by studies and reference projects of human genetic diversity and developments in high-throughput sequencing technologies, genome-wide association studies (GWAS) have been the primary tool to investigate and establish the connection between the genetic data and the observed characteristics or phenotypes, such as occurrence of human diseases or disease biomarkers. For the human genome, the most common form of observed genetic variations are bi-allelic single nucleotide polymorphisms (SNPs) spread across the human genome. The data is usually represented as a numeric matrix where the rows are the genomes from multiple individuals and the columns represent the SNPs. Each bi-allelic SNP (say the ith SNP in the data) can be modeled as a discrete random variable (say, $X_{i}$) taking values from the set {0,1,2}, representing the number of copies of the minor allele in the genome at that position. Also for every individual sample, there is an observed disease phenotype (say, modeled as random variable *Y*), which can be discrete in the simplest case (e.g., presence or absence of a disease), or it can be a continuous valued phenotype (e.g., blood pressure measurements or blood cholesterol values).

Many information theory based approaches have been proposed for disease-gene association studies in the last twenty years. Following a similar characterization as the one proposed by Wu et al. [88], these approaches can be based on: (1) single SNPs; (2) haplotypes; (3) genes; and (4) gene–gene (GXG) interactions. Single SNP based association methods (also known as single-locus tests) are the most common and simplest approaches to identify the disease associated SNPs. Two examples of association tests based on entropy for single SNP-based analysis can be found in [89,90]. These methods devise test statistics treating every SNP as an independent random variable and have low computational complexity proportional linearly to the number of SNPs across the genome considered for the study. However, single SNP based methods suffer from low power to identify causal associations, especially when the SNPs involved are spread across multiples genes [88]. Furthermore, for many such multi-factorial diseases, pathogenic genetic variants usually have a low population of minor allele frequencies (MAFs), making it difficult to identify their effects for relatively small sample sizes [91]. Haplotype-based approaches look at combinations of marker alleles that are closely linked on the same chromosome and tend to be inherited together as a unit, instead of alleles from individual SNPs. This characteristic makes these approaches more suitable to analyze complex multi-factorial disease phenotypes than the single SNP-based methods [88]. Entropy-based tests using haplotypes can be found in [92,93,94]. Their results showed that these tests tend to outperform non-information theoretic tests. However, as pointed out in [88], their main drawback is the computational cost associated with inferring the haplotype phase and frequencies needed for the tests. Closely related to haplotype-based methods are gene-centric approaches that consider genic variants within one gene as testing units and are specially powerful when there is more than one disease variant in a gene. These approaches are specially advantageous when samples come from non-homogenous populations. An entropy-based association test for gene-centric analysis can be found in [95]. In addition to the association test, they also proposed a penalized entropy measure that is used to cluster genotypes and decrease the degrees of freedom of the association test. Gene–gene interaction studies comprise another prominent direction of investigative strategies, those that involve devising test statistics specifically geared towards studying complex, multi-factorial diseases. Many of these proposed methods, discussed below, use information theoretic metrics designed based on Shannon’s entropy. The goal of these information theory based methods is to develop test statistics or scores that measure gene–gene (GxG) statistical interactions (also known as epistasis, [96]) between a set of SNPs and the disease phenotype, and also identify statistical interactions between the genotype and environment (GxE) when environmental variables (factors external to the genome) are included in the study design (Figure 4).

Some of the early information theory based approaches focused on detecting interactions by proposing test statistics based on MI involving two SNPs and the disease phenotype from case-control studies in which a group exhibiting the phenotype/disease is compared to a control group. For example, Fan et al. [97] computed MI between two genetic variants $X_{i}$ and $X_{j}$ separately in cases and controls and used their difference ${\mathrm{MI}}_{cases}\left(X_{i};X_{j}\right)-{\mathrm{MI}}_{controls}\left(X_{i};X_{j}\right)$ as the test statistic to estimate information gain. The idea is to use the MI between the SNPs in the control group as an approximation of the MI in the general population and identify pairs of SNPs with effects substantially bigger than that among the cases. In another proposed method by Yee et al. [98], the difference between the entropy of the phenotype, H(Y), and the conditional entropy of the phenotype given a pair of genetic variants, $H(Y|X_{i},X_{j})$, is normalized with regard to the overall entropy of the phenotype to get the test statistic $\left[H\left(Y\right)-H\left(Y|X_{i},X_{j}\right)\right]/H\left(Y\right)$ that normalizes the information gain with regard to the overall observed entropy of the phenotype. This is also referred to as normalized MI that quantifies the proportion of information contained in the interacting SNPS, $X_{i}$ and $X_{j}$, influencing the phenotype *Y*. In another method, Dong et al. [99] have proposed the ESNP2 (entropy-based SNP–SNP interaction) method integrating two-locus genetic models. Their test statistic is defined as
(17)$$\Delta R_{i,j}=\frac{\min\left(H\left(Y|X_{i}\right),H\left(Y|X_{j}\right)\right)-H\left(Y|X_{i},X_{j}\right)}{\min(H\left(Y|X_{i}\right),H\left(Y|X_{j}\right))}$$
and uses conditional entropy of the phenotype random variable H(Y|.) to measure interaction effects of SNPs $X_{i}$ and $X_{j}$ whenever marginal effects exist. All these methods rely on variations of mutual information (MI) and conditional-MI based two-SNP test metrics and heuristic algorithms of mining the genome looking for two-SNP combinations showing significantly high association scores. A detailed systematic review of these methods, distributional properties of these metrics and their mathematical derivations, and associated algorithms using these information-gain-type quantities for feature selection in the context of genetic analyses can be found in [100].

While the methods summarized above investigate GXG interactions considering relatively simpler two-way SNP–SNP interactions, evidence suggests that higher-order multivariate GxG interactions can contribute to a number of complex traits [101]. Addressing this, several methods have been proposed using more intricate information theory based models that consider the contribution of more than two SNPs to the onset of a phenotype [102,103,104,105,106,107,108,109,110,111,112,113]. For such models, information gain through higher order (with more than two random variables) GxG interactions in explaining the phenotype states as more SNPs are added to a model can be defined from a synergy/redundancy point of view [24,103,104,105,106,108] that quantifies the effects of two or more SNPs compared to their individual effects. The synergy can be positive, when the joint effect of multiple SNPs is larger than the sum of the individual single SNP effects, or negative, when the joint effect is smaller than the sum, indicating information redundancy among SNPs. Some of the prominent methods addressing higher order interactions through synergy and redundancy use multivariate generalizations of MI such as *K*-way Interaction Information (KWII) to parsimoniously model both GXG and GxE interactions and help gain deeper insights into the underlying disease causation pathways by combining them in a single analytical framework [114,115,116,117,118,119,120]. All these methods depend on detection of higher order interactions in terms of synergy/redundancy that relies on robust empirical estimations of metrics such as entropy, MI and KWII. A study by Sucheston et al. [121] systematically evaluated information theoretic metrics such as KWII along with established non-parametric methods, such as Multifactor Dimensionality Reduction [122] and Restricted Partitioning Method [123], comparing power and Type I error. They found that information theoretic models have more flexibility and have excellent power to detect GxG interactions under a variety of conditions including genetic heterogeneity that characterize complex diseases.

Building on similar ideas, epistasis networks proposed by Moore et al. [124] provided a way of using GxG combined with computational network theory. Using the SNPs and their relationships, a network can be created where each node is an SNP and each edge is a KWII between a pair of SNPs in the presence of an observable trait or phenotype, such as bladder cancer susceptibility. An edge is included in the network if the KWII strength is above a threshold that is determined using permutations of the SNP genotypes and phenotype. Once the epistasis network is created, the authors performed network analysis to reveal interesting information. For example, the degree distribution of the vertices in the network was found to closely follow the power law distribution, and hence the network was approximately scale free [125].

In terms of target phenotypes, most of the work described above considers dichotomous phenotypes for case-control studies. Alternative strategies to identify GxE interactions are the case-only and the family-based designs. Case-only studies [126] are used to identify interactions using data from only affected individuals as used in [127,128]. For example, Kang et al. [127] used case-only design to devise a multi-SNP test statistic 2N(1−H(Y))log(W) that is shown to be asymptotically $\chi^{2}$ distributed under the null hypothesis of no association with *W* – 1 degrees of freedom. For a *k*-SNP loci, H(Y) refers to the entropy of the phenotype *Y* obtained using the counts of each unique genotype observed across all the *k* SNPs; *W* is the total number of genotype combinations observed on these loci. These tests are shown to have higher power when compared to the standard $\chi^{2}$ statistical association test. Family-based studies take into account the family relationship of the genomic sequences when comparing them. It considers, for example, that an allele associated with a disease will be transmitted to the affected offspring more often than that expected by chance—the Transmission Disequilibrium Test (TDT) [129]. Examples of such disease association test metrics for family-based study designs can be found in [88,102,130]. Notably, Zhao et al. [130] designed a novel TDT statistic using entropy to generalize the original TDT statistic and incorporated non-linearity in its definition. They demonstrated that the entropy-based TDT test is more powerful than the original TDT test. Besides analyzing binary traits, often GWAS needs to detect gene-disease associations for quantitative traits involving non-discrete real values. Information theoretic methods dealing with such quantitative phenotypes and environmental variables can be found in [116,117,131] that use differential or cross-entropy based generalizations of multivariate test statistics. In a recent review, Galas et al. [132] presented a detailed discussion on the information theoretic formalism for gene association with quantitative phenotypes.

Recently, Tahmasebi et al. used an information theoretic approach to investigate the fundamental limits of GWAS parameters [133]. Their study proposed an abstract probabilistic model to detect the causal subsequence of length *L* for a specific phenotype using a dataset of *N* individuals with genomes of length *G* and their observed characteristics, where the presence of external environmental factors make the relationship between the causal subsequence and the observable characteristics a stochastic function. With increasing value of the model parameters *N*, *G* and *L*, the authors reported observing a threshold effect at (G/N)H(L/G) (*H* denoting binary entropy) that was then used to formulate the capacity of recovering the causal subsequence using information theory. This idea was further extended in a subsequent article [134] by comparing mixed and unmixed populations.

With increasing availability of high-density SNP data from next-generation sequencing techniques like genotyping-by-sequencing, methods for identifying GxG and GxE interactions have rapidly evolved to be an essential technology for association studies. While regression based methods and Bayesian statistics [135,136] have been the primary workhorses for GWAS, information theory has played a crucial role in advancing the field and has been often combined with statistical approaches in reducing dimensionality and devising novel solutions. Nevertheless, there are some open questions that need to be addressed in future research. Although many definitions of statistical interactions and associated test statistics are proposed using information theory, the field would benefit from a unified definition and interpretation of statistical interactions. Additionally, often the underlying distributions of the test statistics under the null and the alternative hypothesis are unknown and more studies are needed as in [137], focusing on investigating the asymptotic behaviors of the estimators involved. Computing higher order test statistics like KWII is computationally expensive as it necessitates entropy computations of all possible subsets of SNP combinations—for example, computing KWII for a set of two SNPs and a disease phenotype entails $2^{3}-1$ entropy computations with three random variables. As a result, computing all possible *k*-SNP combinations in a GWAS study can quickly become computationally infeasible for larger values of *k*. This also increases the number of tests and can reduce power if multiple-testing corrections, such as Bonferroni correction [138], are applied. As a possible remedy, often single SNP analysis is used as pruning step to reduce the number of SNPs to be analyzed prior to interaction analysis, and test significances are estimated through efficient permutation strategies [139]. Another challenge is that the information theoretic test statistics and estimators are, in many instances, still to be adapted to address practical challenges in genetic studies such as accounting for missing genotypes, genotyping errors, phenocopies and genetic heterogeneity. Finally, although some studies exist comparing computational methods for epistatic interaction detection (e.g., [140]), research is still lacking with respect to well-defined studies comparing the powers and performances of several key information theory based as well as other statistical approaches under a variety of experimental conditions. The field will immensely benefit from such systematic studies in the near future.

### 3.5. Protein Sequence, Structure and Interaction Analysis

Proteins are key macromolecules, which mediate their function by interacting with other molecules, including other proteins, nucleic acids, metabolites, and lipids. Major biological processes, such as immunity, metabolism, signaling, gene expression, and molecular machines are controlled through protein interactions [141,142,143]. Due to the critical nature of protein interactions, a key way of investigating an unknown protein is to determine what interactions occur between proteins. The experimental determination of protein–protein interactions (PPIs), using methods such as yeast-two-hybrid, tandem affinity purification, mass spectrometry, DNA and protein microarrays [144,145,146], is time consuming and relatively expensive, and hence, there is a need for accurate computational methods for determining interactions among proteins. With the availability of genomics and proteomics data, several computational methods have been proposed for determination of protein–protein interaction partners. These examples cover different approaches, including Bayesian methods [147], graph-based methods [148], phylogenetics-based methods [149,150], and information theoretic approaches. Applying information theory to protein sequence analysis allows researchers to identify these interactions more easily and without the cost associated with experimental methods. For protein structural analysis, information theoretic methods also offer the advantage of being able to capture relationships at the level of protein families, in contrast to other computational techniques such as PPI structural modeling or docking, which typically operate at the level of individual protein complexes. This is because, information theoretic techniques for protein interaction modeling can capture dependency information across multiple sequences to calculate useful metrics, and are thus able to discern interactions at the protein family level.

An information theoretic approach to infer PPIs is based on the concept of coevolution. The key biological concept that underpins this approach is that interacting proteins adapt together due to evolutionary pressures, which gives rise to relationships that can be observed while observing a large set of data that spans this evolutionary history [151] (such as observing similar gene sequences or gene expression profiles from different cell types or organisms [152]). These relationships can be identified by applying information theoretic concepts such as mutual information (MI). As previously described, MI measures the degree to which random variables are co-dependent. From a multiple sequence alignment of several related proteins, the amino acid distribution at every site can be thought of as a random variable. Following this, MI can quantify the degree of reduction of uncertainty at one site, based on the knowledge of the distribution at a different site. In a notable initial application of these approaches, Giraud et al. [153] applied the concept of coevolution to proteins in their study in 1998. Since that time, several attempts have been made to refine this measure for their applicability to proteins. Several studies [154,155,156] have described approaches to model the effect of phylogenetic relationships between the proteins and separated this from the coevolution between pairs of sites within a multiple sequence alignment, and have demonstrated that these techniques are able to improve the applicability of such metrics to identifying PPIs. Figure 5 shows the role of information theory in a typical coevolution approach.

These methods also have a significant impact in the field of structural biology. A key element to understanding and predicting the function of a protein is its structure. While a structure is largely determined by its sequence, over the years, structures for several proteins have been determined by X-ray crystallography, Nuclear Magnetic Resonance (NMR) methods, and more recently Cryo-Electron Microscopy (CryoEM). As the number of structures solved and deposited in the Protein Data Bank (PDB) keeps increasing, the vast explosion in data has allowed the application of information theoretic approaches to play impactful roles in structural biology [157].

Since complementarity is a key feature of three-dimensional protein interactions, coevolution would suggest that when a substitution occurs, a compensatory substitution would occur in an interacting partner, that would be captured by the metric regardless of where in the sequence the mutation occurred. Validation of coevolution-based approaches discussed previously has been performed by studying the structures. Specific examples encompass a variety of applications, some of which have been described in a study by Little et al. [10]. This cross-functional work includes a study of coevolution in the PDZ domain of human protein Erbin, also among related, previously known and annotated functional domains from the PFAM database, as well as across known catalytic sites. Other studies have also demonstrated the immense potential that coevolution offers to structural and systems biology. Specialized approaches for modeling protein coevolution are required since amino acids have several interacting partners, and MI based metrics capture interactions that may be due to indirect coupling [157]. This additional complexity is resolved in methods such as DCA [158] or GREMLIN [159], which are able to highlight direct interactions and remove the noise from indirect interactions.

In a recent study, Cong et al. [160] have applied these techniques to large-scale data from proteomics studies. This study investigated coevolution between 5.4 million pairs of proteins in *Escherichia coli* and between 3.9 million pairs in *Mycobacterium tuberculosis*, and reported that in conjunction with structural modeling, they were able to predict PPIs with an accuracy much higher than that found by traditional proteome-wide two-hybrid screens. This study only highlights the potential for information theory methods to go beyond traditional methods and help identify PPIs for unexplored organisms.

Not addressed here, but other equally challenging and interesting uses of information theory to analyze PPIs are addressed in a review article by Mousavian et al. [5]. In that review, the authors discuss in detail applications of information theory in dealing with PPI networks, describing a variety of applications such as protein complex identification, network complexity analysis, finding subnetwork markers, and other questions addressed by previous studies applying information theory for PPIs. In contrast, for this review, we have highlighted some studies that build on the concept of sequence coevolution and discuss the future scope of these methods in solving unsolved problems. As these methods are further refined and integrated with structural biology, as in Cong et al. [160], they can be further improved with the goal to map all cellular PPIs, enabling discovery of the “interactome”—the cellular protein interaction network. This represents a challenging task of evolving from the current paradigm of discovering interactions between specific proteins or protein families to mapping at the scale of millions of proteins. Other exciting and emerging applications in the structural domain include recent studies of specificity in these interactions and how they can help guide in-silico assemblies of protein complexes [157].

### 3.6. Metabolic Networks and Metabolomics

Metabolic networks are networks that describe the physiological and biochemical properties of a cell. The components are metabolites, chemical reactions that transform the metabolites to each other catalyzed by metabolic enzymes, metabolic pathways, as well as the regulatory interactions that guide these reactions. Unlike other types of systems biology networks, studies of metabolic networks have been limited due to difficulties in generating large amounts of metabolite concentration data and lack of knowledge about the kinetic properties of the involved reactions. The recent rapid evolution of metabolomics based on mass spectrometry and nuclear magnetic resonance has made possible faster and more in-depth studies of metabolic networks. In this section, we will introduce some studies that have applied the concepts of information theory in understanding metabolic networks. An overview illustrating some of the approaches discussed is presented in Figure 6.

Reconstruction of metabolic networks remains a central topic similar to other types of biological networks. A top-down approach to metabolic network reconstruction is by reverse engineering of metabolome data. Popular methods for inference of regulatory networks, ARACNE [37] and CLR [36] have been applied to reconstruct cellular metabolic networks [161]. Both these methods leverage mutual information (MI). A method based on conditional MI has also been used [162]. These methods can detect non-linear correlations compared to simpler correlation-based relatedness scores. Saccenti et al. [163] proposed a “wisdom of crowds” approach that considers the consensus obtained from four different approaches, ARACNE, CLR, PCLRC (Probabilistic Context Likelihood of Relatedness on Correlations) [164] and Pearson correlation.

Molecular networks built from metabolite profiling can exhibit a large degree of diversity across individuals and this variability reflects the intrinsic diversity observed among the individual metabolic phenotypes. To characterize this diversity, the concept of entropy can be used [163]. Importantly, both entropy profiles of single metabolites and entropies of one metabolite relative to others that characterize dependencies and correlations between metabolites in a network context need to be considered. In the recent review published by Everett et al. [165], metabolic entropy is considered as one of the four fundamental approaches to the generation and utilization of metabotype data for metabolic phenotyping in diagnosis and prognosis, another being the metabolic network itself.

An important goal of metabolic network studies is understanding the dynamical behaviors of metabolic networks and the functions generated by them. Despite their topological complexity, metabolic networks avoid complex dynamics and maintain a steady-state behavior. A combination of the Shannon entropy and the word entropy [166] capable of separating different dynamic regimes in metabolic networks have been used to reveal that this pronounced regularization of dynamics is encoded in the network topology [167]. Nykter et al. [168] also studied network structure-dynamics relationships, using Kolmogorov complexity as a measurement of information distance between pairs of network structures and between their associated dynamic state trajectories. The possible dynamics of metabolic networks was studied in Grimbs et al. [169] by using a kinetic model. The enzyme kinetic parameter space was sampled and the metabolic dynamic states were evaluated statistically. MI was then used as one of the three distinct measures to assess the relative impact of kinetic parameters on the stability and robustness of metabolic networks. Cellular metabolic systems form self-assembled aggregates and the activities of cellular enzymes can also exhibit spontaneous spatial-temporal functional structures. Entropy is a useful concept in the study of these dynamical systems [11]. In particular, Kolmogorov–Sinai entropy, which can be estimated from a finite number of observations using a family of statistics named Approximate Entropy (ApEn), provides a good measure of the complexity and information for the study of attractors in biochemical systems.

Metabolic networks are often modeled and simulated using Flux Balance Analysis (FBA) in order to study the physiology of the relevant microorganism or cell. FBA can predict metabolic reaction rates, also known as fluxes, without using kinetic parameters, by representing a metabolic network in the form of a set of mass balance equations based on the stoichiometry of each reaction, and computing reaction fluxes to match biomass production rate to a measured growth rate. However, substantial cell-to-cell growth rate fluctuations exist even in well-controlled steady-state conditions. De Martino et al. [170] introduced a generalization of FBA to the single-cell level based on the maximum entropy principle [171]. The idea is to look for an as-random-as-possible distribution over fluxes that is matching the experimentally measured average growth rate. This maximum entropy metabolic modeling has been shown to provide a better match to experimentally measured fluxes and it makes a wide range of predictions such as on flux variability, regulation, and correlations.

Finally, metabolic networks transform nutrients into biomass and it is of interest to understand how a cell acquires information on nutrient availability through nutrient sensing and how a metabolic network uses this information. Wagner at al. [172] proposed a way to relate the nutrient information to the error in cell’s measurement of nutrient concentration in the environment. FBA was then used to show that nutrient sensing inaccuracy is translated logarithmically into reduced cell growth, and for microbes like yeast, cells would need to estimate nutrient concentrations to very high accuracy to ensure optimal growth.

### 3.7. Connections to Optimization and Dimensionality Reduction in Biology

In this section, we briefly review some key applications connecting information theory with optimization, notably through Maximum Entropy Production (MEP) and then discuss information theory in the context of dimensionality reduction for omics analysis.

#### 3.7.1. Optimization in Biology

Natural selection can be considered an optimization process itself, given environmental constraints, leading to “survival of the fittest”. While mathematical models have been used to model biological processes ranging from microbial metabolism [173,174] to plant growth [175,176] to human disease [177,178], Maximum Entropy Production (MEP) has served as an ultimate measure of fitness by “survival of the likeliest” [179,180]. MEP is defined by [180] as a memoryfull (or path-dependent) change in a system’s state probability distribution. While a memoryless entropy change between distributions *P* and *Q* is often measured by Kullback–Leibler Divergence (KLD) in physics, MEP, as applied to a biological system, reflects the thermodynamic likelihood of all chemical reactions during a process [180]. In plants, for example, mechanistic models have successfully predicted specific, observable traits such as plant height or processes like evapotransporation [175]. These models are then used as proxies for optimization targets like disease resistance and growth rates. In the early 2000s to 2010s, models were developed by [179] and others, which assumed fitness in any environment is best described by Entropy Production. While the MEP principle cannot be proved [181], it has been demonstrated as practical and applicable for both individual organisms and entire ecosystems [179,181,182]. More recently, Cannon et al. [180] used the MEP principle to model central metabolism in the fungus *Neurospora crassa* and used it to estimate kinetic rate constants normally obtained using painstaking experimentation [183]. This optimization objective used the MEP principle by choosing the parameters (inferred rate constants) that “can happen in the greatest number of ways” [180]. A hypothetical search space for such an optimization is depicted in Figure 7.

#### 3.7.2. Dimensionality Reduction for Omics Analysis

Omics experiments suffer from the small-*n*-large-*p* problem, where a relatively small number of samples, *n*, each comprise a large number of colinear variables, *p*. Dimensionality reduction, reviewed by Sorzano et al. [184], amounts to the pursuit of a projection to lower-dimensional space that is *Efficient*, *Relevant*, and *Meaningful*. *Efficient* means the projection space has few dimensions and possibly weighs few of the *n* input variables (sparse or regularized), *Relevant* means that it maintains information related to the experiment, and *Meaningful* means that the projection is oriented in a useful or interesting position. While Principle Components Analysis (PCA) [185] finds the *Meaning* by projecting data to orthogonal axes ordered by their variance, Independent Component Analysis (ICA) [186] explicitly uses information theory to find *Meaning* differently. In ICA, the latent factors are assumed to be statistically independent and requires non-normality in the distributions of the independent latent factors. Reviewed in [187], this non-normality requirement is crucial to ICA and is optimized in different ways using different calculations of non-normality. One of those measures is *Negentropy,*
*J*,
(18)$$\begin{array}{c} J(Y)=H(Z)-H(Y) \end{array}$$
which is a measure of how different the differential (continuous) entropy *H* (Equation (Equation 4)) of a random variable *Y* is from that of a truly normal random variable *Z* with the same variance-covariance structure. The reason H(Y) is subtracted from H(Z) is given a fixed second moment (variance), the normal density has the greatest entropy. Further, minimum MI between projection axes is a good measure of maximum *Negetropy* [187]. An advantage of ICA is that latent factors need not be orthogonal, as shown in Figure 8. ICA is applied not only for dimensionality reduction, but also in signal processing [187]. A difference between the two is that ICA does not rank latent factors in order of importance, as PCA does. Lastly, ICA calculation is not deterministic, and can sometimes lead to unstable results.

## 4. Discussion

In this article, we have performed a broad overview of applications of information theory in many key areas within the gamut of computational biology with focus on more recent developments (summarized in Table 1 and Table 2). Information is intrinsically central to biology, most obviously because genetic information is stored in the DNA. Since the seminal work done by Shannon over seventy years ago, information theory with its foundations in statistical mechanics and communication theory, has made a tremendous impact to computational biology. Not only has it provided many ways of parsimoniously modeling and capturing non-linear associations between key components in biology (such as gene expression, DNA nucleotides, protein residues), it also helped represent dynamic biological systems as stochastic random processes. Its popularity also stems from the fact that information theory provides a strong alternative to many conventional statistical approaches often challenged with complex parameterization and computational intractability due to high dimensionality of input data. At the same time, although information theoretic applications have grown at a very fast pace and information theory based theoretical frameworks can be adapted to a wide range of problems, many challenges exist that should be considered prior to any analysis, as discussed in the context of biological applications in Andrews et al. [188] and for general data mining by Holzinger et al. [189]. The central concept in information theory is Shannon’s entropy that is based on expected value of a probability distribution akin to statistical averaging. Because heterogeneity of traits and phenotypes is a rule in biology, rather than an exception, averages may not always work well in explaining or predicting behaviors. Many proposed information theory based test statistics should be used with caution as stationarity assumptions generally do not hold, sample sizes may be too small to support the law of large numbers and asymptotic properties are not well understood. Estimation of metrics such as interaction information relies on robust empirical estimations of multivariate distributions and joint probability mass functions with groups of random variables that are not straightforward at all, and more so, when data points are limited. Difficulties related to obtaining sufficient sample sizes and the computational burden associated with such estimations using high-dimensional and heterogeneous data often encountered in biology can result in bottlenecks in the application of information theory to systems biology [190]. Another set of issues to be tackled exist in the modeling front that include behavior of noise and robustness of models to imperfections and irregularities, a frequent occurrence in the experimental biological domain [188,191]. Assumptions of ideal conditions (infinite block lengths, additive white Gaussian noise, i.i.d distributions, etc.) underlying many classical information theoretic results may not always hold when the conditions are relaxed. Finally, some other open questions are how to select an appropriate entropy measure, its parameters and higher order metric(s) using the entropy measure to address a particular problem in biology and how to generally benchmark multiple such entropy measures [189]. More research is required addressing the above-mentioned challenges within specific applications in biological domains.

In the past few years, cutting-edge and high throughput technologies and experimental capabilities in biology have enabled rapid collection of unprecedented amounts of data ranging from millions of genomic sequences, images of physiological structures, high resolution microscopy images of cell morphologies to petabytes of health records. Therefore, it is worthwhile to briefly mention the role of information theory in the context of the recent mega-trends in data generation and analytics, particularly with respect to the newer paradigms of “Big Data” and “Deep Learning” that have emerged over the past two decades. The 3V’s (“Volume”, “Velocity” and “Variety”) constitute three key properties of any “Big Data” in general that informally refers to the data deluge that generates large “Volumes” of data at high “Velocities” with a lot of “Variety” [192,193]. Within biology, genomics, through production of high-throughput sequencing data, continues to lead in terms of data growth and availability [192,194]. Information theoretic approaches, being at the heart of some of the popular pattern mining and statistical machine learning methods, stand to benefit from the increasing “Volume” and “Velocity” primarily through more reliable estimations of underlying empirical data distributions leading to improved estimation of metrics such as MI and KWII. However, ever increasing production of high-throughput biological data poses serious challenges to the conventional solutions for storing, processing and transmitting these data. Consequently, in addition to dimensionality reduction strategies for omics analysis, DNA based data storage [72] and data compression methods [195] for various biological data types have become active areas of research over the last decade. Because of the long-term stability and information density, using DNA as an archival medium is on track to become an appealing medium for handling next generation data “Volume”. Multiple methods have been proposed in the last few years [72,77,78,85,86,87] that aim to harness the properties of DNA to mathematically model it as a storage channel and improve it’s information capacity. In the data compression area, some of the early compression methods, such as GenCompress [196] used reference genomes to map short sequences and then used entropy coding algorithms to encode the addresses of the short sequences, their lengths and their probable substitutions. More recent methods like CoGI [197] and iDoComp [198] can be used for both reference-free as well as for reference-based genome compression and rely on applying advanced data structures and algorithms such as suffix arrays and rectangular partition coding [199] to reduce mapping size before using entropy encoding in the final compression step. Another promising direction to manage this data deluge can be found in the work of Yu et al. [200] and later advanced by Ishaq et al. [201]. Their research proposed information theory and hierarchical clustering based framework for similarity search in massive biological datasets based on characterizing a dataset’s entropy and fractal dimensions, and enabled reduction in data volume by pruning redundant data while preserving the essential structures and patterns within the data. This not only attempts to address data “Volume”, it also holds the potential to improve data “Veracity” (from 5V categorization of “Big data” [193]) that reduces data redundancy and noise and can lead to improved estimations of information theoretic as well as statistical metrics in downstream applications. Fueled by the ever-increasing growth in biological data and emerging techniques such as CRISPR based genome-editing, we are only beginning to explore the capacity of information science to advance research in biological data compression and DNA based storage, and we expect to see many more exciting developments in these areas.

More recently, inspired by knowledge of neural information processing and functioning of the brain, and powered by availability of modern powerful GPUs, artificial intelligence and deep learning have made impressive advances in numerous applications ranging from computer vision, natural-language and speech processing to bioinformatics and computational biology. Keeping up with this trend, information theory, in many forms, has now become ubiquitous within many state-of-the-art algorithms in machine learning and deep learning algorithms. It has helped advance theoretical developments of optimization and training in deep learning, that drives many emerging applications in biological data science such as genomic predictions and imaging genomics exploring relationships between genotypes, phenotypes and clinical outcomes [205,206]. For example, cross entropy has become a standard for comparing two probability distributions and is a popular loss function for deep neural networks, in both binomial and multinomial classification scenarios. Information theory is also continuing to play a central role in on-going research investigating information bottleneck and related principles in the analysis and design of representation learning and optimization algorithms for training deep neural networks, for example, as in [207].

Lastly, with recent advancements in quantum computations, Shannon’s classical information theory is paving the way for using quantum information theory with physics to further the understanding of biological systems. Active research is being pursued in many uses of quantum information theory, such as developing quantum biological channel models suitable for the study of quantum information transfer from DNA to proteins [208], quantum-mechanical modeling of spontaneous, induced, and adaptive mutations and their role in cancerous tumor developments [209] and using both classical and quantum error-correction coding in genetics and evolution [210]. We conjecture that information theory has a key role to play in many theoretical and application developments in these areas in the near future.

## Figures and Tables

**Figure 1 entropy-22-00627-f001:**
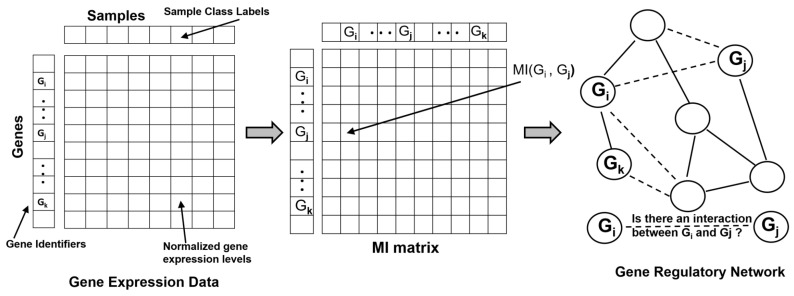
Schematic showing the Gene-regulatory Networks (GRN) reconstruction problem where undirected edges are inferred using information theoretic methods.

**Figure 2 entropy-22-00627-f002:**
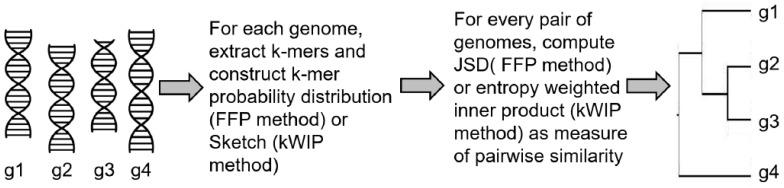
Schematic showing an overview of alignment-free sequence comparison.

**Figure 3 entropy-22-00627-f003:**
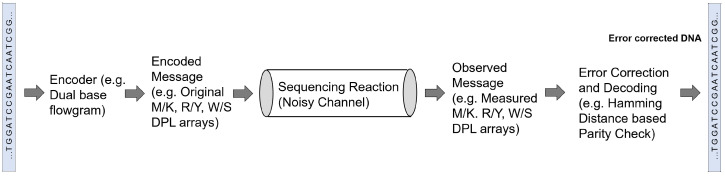
Illustration of an information theoretic communication model inspired representation for error correction in genomic sequencing based on the work of Chen et al. [65].

**Figure 4 entropy-22-00627-f004:**
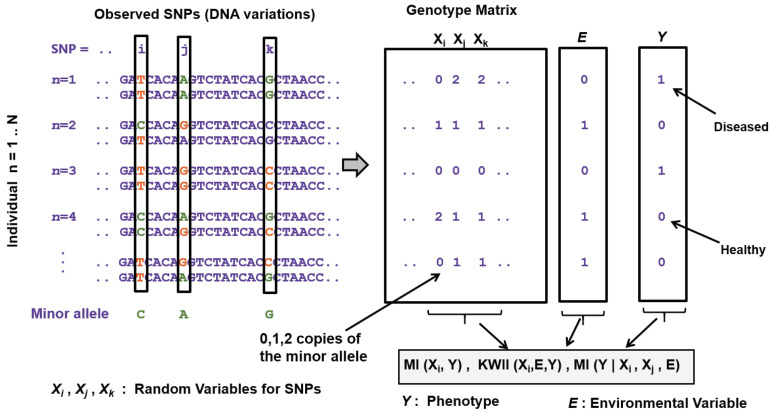
Overview of the genome-disease association problem and some example information theoretic metrics calculated for identifying GxG and GxE, a binary phenotype and environmental variable is shown for simplicity.

**Figure 5 entropy-22-00627-f005:**
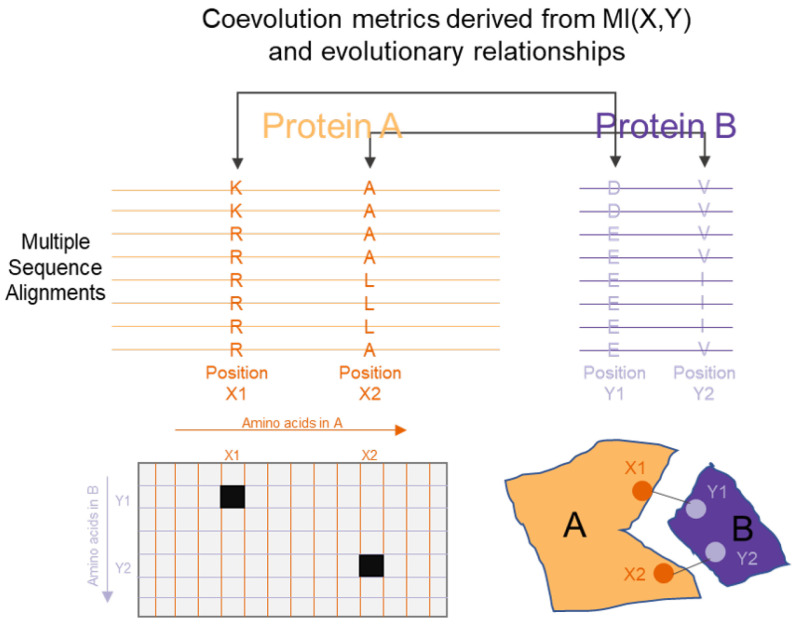
Simplified illustration of how mutual information (MI) can be used to capture links between two proteins from their Multiple sequence Alignments. High MI between 2 columns of the alignments are associated with interactions in their structures.

**Figure 6 entropy-22-00627-f006:**
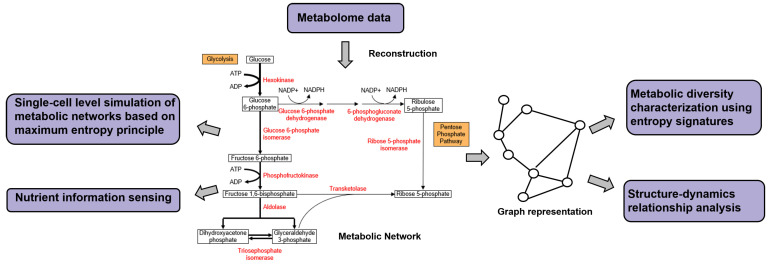
An example of metabolic networks. Nodes represent metabolites and are transformed into each other through chemical reactions catalyzed by metabolic enzymes (red). Metabolic pathways are formed by a linked series of chemical reactions that collectively perform a biological function. Reconstruction of metabolic networks is typically done by reverse engineering of metabolome data, where information theoretic methods ARACNE and CLR have been applied. Reaction fluxes depicted by the width of the reaction arrows can be predicted using FBA. Also shown are examples of downstream analysis of a metabolic network using information theory for metabolic diversity characterization, structure-dynamic relationship analysis, single-cell level simulation of metabolic networks with FBA and nutrient information sensing.

**Figure 7 entropy-22-00627-f007:**
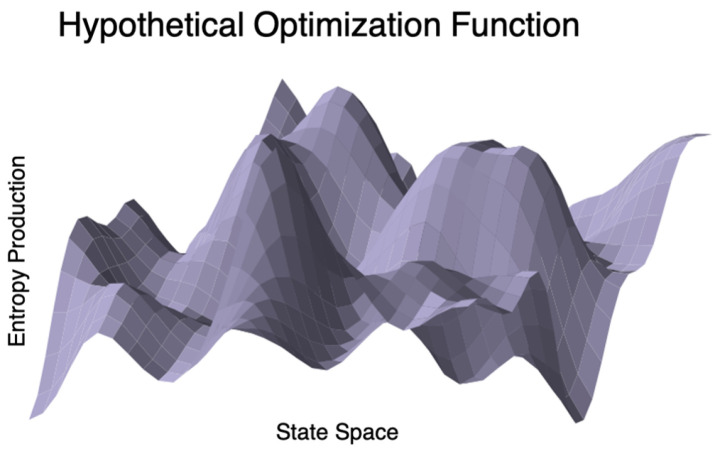
Biology presents many optimization problems. Shown here is a hypothetical search space that could represent anything from an organism’s traits to experiment designs to visualization parameters. In these cases, the vertical axis could be fitness/MEP, experiment value, and network edge crossovers, respectively.

**Figure 8 entropy-22-00627-f008:**
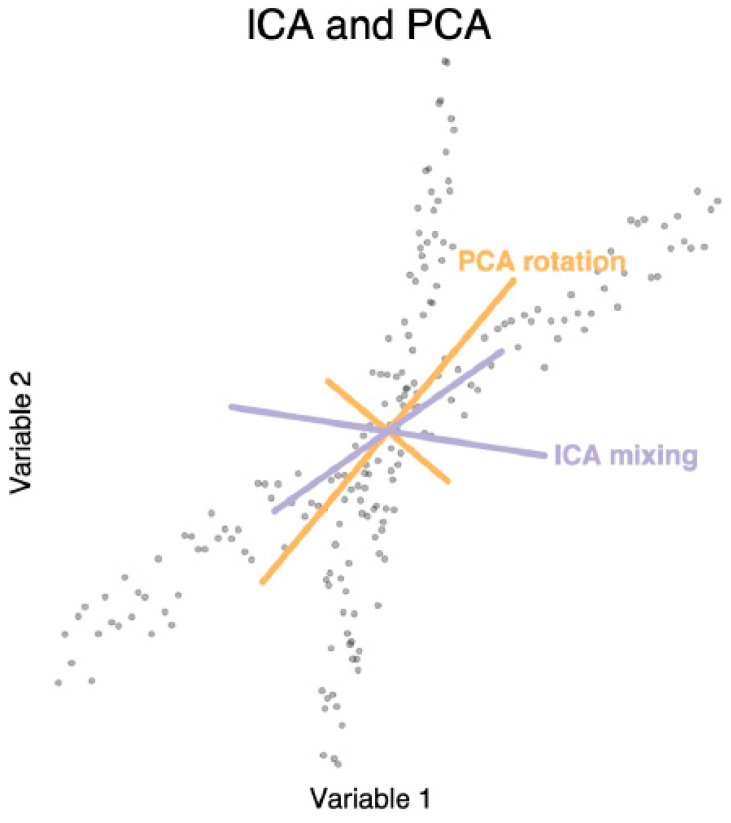
Independent Components Analysis (ICA) can compute latent factors in data like Principle Components Analysis (PCA). ICA is not limited to orthogonal mixtures like PCA, as shown in this simulated dataset. The ICA mixing vectors align to the data better than the PCA rotation vectors.

**Table 1 entropy-22-00627-t001:** Summary of key information theoretic methods for areas discussed in Section 3.1, Section 3.2, Section 3.3 and Section 3.4.

Area	Information Theoretic Methods	How Information Theory Is Used
Reconstructing Gene Regulatory Networks	Relevance Networks [34]	Used MI larger than a given threshold to construct GRNs
	CLR [36]	Used MI to construct GRN, filters spurious edges by estimating its background distribution
	ARACNE and its extensions [37,38,39,40]	Used MI to construct GRN, filters edges using null distribution and DPI, higher order DPI to improve inference performance, adaptive binning strategy to estimate MI efficiently
	PIDC [27]	GRN constructed using PID to explore dependencies between triplets of genes in single-cell gene expression datasets
Alignment-free phylogeny	FFP [54]	Calculated JSD as pairwise distances between two genomes using normalized k-mer frequencies
	kWIP [55]	Constructed a sketch data structure using all k-mers from a genomic sequence, computed inner product between the two sketches weighted by their Shannon’s entropy across the given dataset
Sequencing and Error Correction	Motahari et al. [66]	Used Renyi entropy of order 2 to find the minimum fragment length and coverage depth needed for the assembling reads to reconstruct the original DNA sequence with a given reliability
	Chen et al. [65]	Analyzed the information redundancy in dual-base degenerate sequencing by comparing entropy information content of multiple DPL (degenerate polymer length) arrays
	Anavy et al. [77]	Proposed encoding and decoding for composite DNA based storage and error correction through developing composite DNA alphabets and using KLD to select the best alphabet model
	Choi et al. [77]	Used eleven degenerate bases as encoding characters in addition to ACGT to increase information capacity limit and reduce the cost of DNA per unit data
Genome-wide disease-gene association mapping	Fan et al. [97], Yee et al. [98], Dong et al. [99]	Proposed test statistics based on MI and conditional entropy involving two SNPs and the disease phenotype from case-control studies
	Varadan et al. [103], Anastassiou [104], Curk at al. [105], Hu et al. [106,108]	Used synergy to analyze GXG statistical interactions
	Chanda et al. [24,114,115,118], Tritchler et al. [120]	Used multivariate information theoretic metrics and higher order models (e.g., KWII) to analyze statistical GXG and GxE interactions
	Moore et al. [124]	Used KWII to represent edges in epistasis networks
	Tahmasebi et al. [133]	Used entropy to formulate the capacity of recovering the causal subsequence
	Chanda et al. [116], Knights et al. [117], Yee et al. [131], Galas et al. [132]	Discussion and analysis of Information theoretic methods dealing with quantitative phenotypes and environmental variables
	Andrade et al. [128], Kang et al. [127]	Developed information theoretic test statistics for single-group or case-only studies.
	Tzeng et al. [92], Zhao et al. [93,94]	Developed entropy-based tests using haplotypes
	Cui et al. [95]	Developed entropy-based association test for gene-centric analysis considering variants within one gene as testing units
	Zhao et al. [130], Brunel et al. [102], Wu et al. [88]	Designed and discussed entropy based disease association test metrics for family-based studies

**Table 2 entropy-22-00627-t002:** Summary of key information theoretic methods for areas discussed in Section 3.5, Section 3.6 and Section 3.7.

Area	Information Theoretic Methods	How Information Theory Is Used
Reconstruction and analysis of Metabolic Networks	CLR [36], ARACNE [37], PCLRC [164]	Uses MI to construct metabolic networks from metabolite concentration data, filters spurious edges by estimating its background distribution [36,164] or DPI [37]
	Marr. et al. [166,167]	Network analysis of metabolic networks using Shannon and word entropy to reveal regularization dynamics encoded in network topology
	Nykter et al. [168]	Studied network structure-dynamics relationships, using Kolmogorov complexity as a measure of distance between pairs of network structures and between their associated dynamic state trajectories
	Grimbs et al. [169]	Stoichiometric analysis to parameterize the metabolic states, assessed the effect of enzyme-kinetic parameters on the stability properties of a metabolic state using MI and Kolmogorov–Smirnov test
	Fuente et al. [11].	Studied properties of dissipative metabolic structures at different organizational levels using entropy
	De Martino et al. [170]	Introduced a generalization of FBA to single-cell level based on maximum entropy principle
	Saccenti et al. [163]	Investigated the associations and the interconnections among different metabolites by means of network modeling using maximum entropy ensemble null model
	Wagner et al. [172]	Proposed an information theoretic way to relate the nutrient information to the error in a cell’s measurement of nutrient concentration in its environment
Protein interaction analysis	Wollenberg et al. [154], Tillier et al. [155], Dunn et al. [156], Kamisetty et al. [159], Morcos et al. [158]	Mutual Information combined with evolutionary information and refined with structural information to identify protein interactions
Optimization, Dimensionality Reduction	Cannon et al. [202,203], Thomas et al. [204]	Used MEP to simulate central metabolism in the fungus Neurospora crassa [180], tricarboxylic acid cycle model optimization in microbes [204].
	Hyvarinen et al. [187], Comon et al. [186]	Used negentropy and minimization of MI to obtain the components in ICA

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
