# Peer review of "Information Theory in Computational Biology: Where We Stand Today"

_entropy, 2020, doi:10.3390/e22060627_

Round 1

Reviewer 1 Report

The authors reviewed the use of information theory in various fields of computational biology. Some sections of the review are little superficial (e.g. genome reconstruction, error correction, GWAS) and include tangentially related information (e.g. simulated annealing, PCA). This review would benefit from the removal of unrelated and unnecessary information, a deeper discussion of the improvement that information theory brings over alternative approaches, as well as a clearer discussion of the specific challenges that remain to be solved in each field.

Specific comments

Line 54. Omit "exhaustive" to avoid overclaiming.

Line 104. "on an average" -> "on average"

Line 105. Remove "the distribution of events X = x_i is very deterministic and". The word "deterministic" is conflated with "concentrated on particular values." The second part of the clause is sufficient to convey the intended meaning.

Equation 8. The bounds of integration should not be assumed to be (-Inf, Inf). The integration should be over the domain of X.

Line 668. "comprehensive" is an overstatement.

Line 200. The limitation regarding the direct vs. indirect relationship should be mentioned earlier in the paragraph.

Line 224. I(G_i; G_j) is not defined.

Line 217. It is unclear how the PID method addresses zero-inflated, single-cell RNAseq data.

Line 227-230. Re-word the sentence beginning with "This allows..."

Line 237 "methods for inferring GRNs are often specific to expression data from certain cell types". The meaning here is unclear. Additionally, there should be some discussion about how the methods are benchmarked or evaluated, and what constitutes the ground truth.

Line 254. "violated in the real word" -> "violated in biological genomes due to rearrangement events"

Line 314. Read lengths can be 100-10000. Oxford Nanopore and PacBio reads can reach 10kbp or greater.

Section 3.3. It is unclear how the genome reconstruction studies mentioned is related to or uses information theory. It is also unclear how the error-correction algorithm uses information theory.

Line 369. Omit "Once DNA sequencing is completed". DNA sequencing is usually not used for GWAS studies. There is no need to further propagate the confusion between sequencing and genotyping. Perhaps start with "A central challenge in genetics..."

Line 457. Omit "With increasing number of human genomes being sequeneced." Same concern as before.

Section 3.4 reads more like an introduction to GWAS rather than a review of the use of information theory in GWAS.

Line 478. "bayesian" -> "Bayesian"

Line 480. Omit the sentencing beginning with "in this review subsection". This sentence is both problematic and redundant.

Line 623. It is unclear why simulated annealing is relevant to the topic of the review.

Line 635. Omit the phrase "probably derived from ..."

Line 664-665. Re-word sentence.

Line 694. The challenges referenced here should be discussed in the relevant sections.

Inline citation should be consistent throughout the manscript (authors in [X] vs. X et al. vs. X and colleagues).

Reviewer 2 Report

The article entitled “Information theory in computational biology: where we stand today” by Chanda et al. is fascinating and relevant to the current big data-related challenges in the area of computational biology. However, the article is lacking vital information/some explanations and graphical representations. The following comments might help the authors to make this article more informative and understandable to experts and non-expert in the field of information theory:

MAJOR COMMENTS:

The authors did not mention any drawbacks or challenges in using information theory in computational biology. The authors should either mention these challenges in the review or cite the relevant article(s). The “discussion” section is a good place to mention these challenges and possible ways for future directions. The field of Information theory constantly talks about the 3Vs (Volume, Veracity and Velocity) of Big Data. These concepts also apply to Biological Data. It would be interesting to have a short paragraph of how we can apply IT concepts in handling these 3Vs for biology. Furthermore, the manuscript does not mention any comparative analysis of applications that use entropy concepts and those that do not.

Section 2:

- Please incorporate examples of biological molecules or processes where these concepts (e.g., entropy, relative entropy, condition entropy, and so on) are relevant. It will help non-experts (in the field of information theory) to understand the connections between reviewed concepts and their biological applications. This will make it even easier to follow the details of the application of these approaches in the second part of the review. 

- Sections 2.4. Mutual Information, and 2.5. Interaction Information are impressive, however, please include more explanations in both sections - what could these random biological variables be. Also, in 2.5. Interaction Information, do more random variables result in any challenges such as computational time, increased errors in KWII values? IT constantly deals with data compression for data analysis and data transfer. Biological data analysis, for example GWAS data, is extremely time consuming, and computationally expensive. There is not mention of how it is possible to use IT to compress these data for data comparison and data transfer. Please include a detailed discussion.

Section 3:

- This section is written well and very informative. However, graphical representations of each application are missing, which makes this section difficult to understand for non-experts and some potential end-users of the proposed concepts (biologists, healthcare personnel). Therefore, adding a graphical illustration in each sub-section is recommended. 

- Also, the authors should add a table, including a list of information theory-based tools currently being used with their specific features and suitable references.

MINOR COMMENTS:

- Lines 31-33: Please provide reference(s).

- Line33-38: This sentence is too long. Please consider breaking into two or more. Also, it is recommended to check such sentences throughout the text.

- Section 2.4: References are missing.

Round 2

Reviewer 1 Report

The authors have adequately addressed my concerns.

Reviewer 2 Report

The authors addressed all the comments and improved the manuscript significantly. Please see below some minor comments which can be incorporated into the article.

  1. Figure 1: What do Gs and MI denote? Please mention in the figure caption. Also, what is the difference between the line and the dotted line in the network?
  2. Figure 3: Describe M/K, R/Y, W/S, and other abbreviations in the figure caption.